# Batched Gaussian Process Bandit Optimization via Determinantal Point Processes

**Tarun Kathuria, Amit Deshpande, Pushmeet Kohli**
Microsoft Research
t-takat@microsoft.com, amitdesh@microsoft.com, pkohli@microsoft.com

## Abstract

Gaussian Process bandit optimization has emerged as a powerful tool for optimizing noisy black box functions. One example in machine learning is hyper-parameter optimization where each evaluation of the target function may require training a model which may involve days or even weeks of computation. Most methods for this so-called "Bayesian optimization" only allow sequential exploration of the parameter space. However, it is often desirable to propose *batches* or sets of parameter values to explore simultaneously, especially when there are large parallel processing facilities at our disposal. Batch methods require modeling the interaction between the different evaluations in the batch, which can be expensive in complex scenarios. In this paper, we propose a new approach for parallelizing Bayesian optimization by modeling the diversity of a batch via Determinantal point processes (DPPs) whose kernels are learned *automatically*. This allows us to generalize a previous result as well as prove better regret bounds based on DPP sampling. Our experiments on a variety of synthetic and real-world robotics and hyper-parameter optimization tasks indicate that our DPP-based methods, especially those based on DPP sampling, outperform state-of-the-art methods.

## 1 Introduction

The optimization of an unknown function based on noisy observations is a fundamental problem in various real world domains, e.g., engineering design [33], finance [36] and hyper-parameter optimization [29]. In recent years, an increasingly popular direction has been to model smoothness assumptions about the function via a Gaussian Process (GP), which provides an easy way to compute the posterior distribution of the unknown function, and thereby uncertainty estimates that help to decide where to evaluate the function next, in search of an optima. This *Bayesian optimization* (BO) framework has received considerable attention in tuning of hyper-parameters for complex models and algorithms in Machine Learning, Robotics and Computer Vision [16, 31, 29, 12].

Apart from a few notable exceptions [9, 8, 11], most methods for Bayesian optimization work by exploring one parameter value at a time. However, in many applications, it may be possible and, moreover, desirable to run multiple function evaluations in parallel. A case in point is when the underlying function corresponds to a laboratory experiment where multiple experimental setups are available or when the underlying function is the result of a costly computer simulation and multiple simulations can be run across different processors in parallel. By parallelizing the experiments, substantially more information can be gathered in the same time-frame; however, future actions must be chosen without the benefit of intermediate results. One might conceptualize these problems as choosing "batches" of experiments to run simultaneously. The key challenge is to assemble batches (out of a combinatorially large set of batches) of experiments that both explore the function and exploit by focusing on regions with high estimated value.

**Our Contributions** Given that functions sampled from GPs usually have some degree of smoothness, in the so-called *batch Bayesian optimization* (BBO) methods, it is desirable to choose batches which are diverse. Indeed, this is the motivation behind many popular BBO methods like the BUCB [9], UCB-PE [8] and Local Penalization [11]. Motivated by this long line of work in BBO, we propose

a new approach that employs Determinantal Point Processes (DPPs) to select diverse batches of evaluations. DPPs are probability measures over subsets of a ground set that promote diversity, have applications in statistical physics and random matrix theory [28, 21], and have efficient sampling algorithms [17, 18]. The two main ways for fixed cardinality subset selection via DPPs are that of choosing the subset which maximizes the determinant [DPP-MAX, Theorem 3.3] and sampling a subset according to the determinantal probability measure [DPP-SAMPLE, Theorem 3.4]. Following UCB-PE [8], our methods also choose the first point via an acquisition function, and then the rest of the points are selected from a relevance region using a DPP. Since DPPs crucially depend on the choice of the DPP kernel, it is important to choose the right kernel. Our method allows the kernel to change across iterations and automatically compute it based on the observed data. This kernel is intimately linked to the GP kernel used to model the function; it is in fact exactly the posterior kernel function of the GP. The acquisition functions we consider are EST [34], a recently proposed sequential MAP-estimate based Bayesian optimization algorithm with regret bounds independent of the size of the domain, and UCB [30]. In fact, we show that UCB-PE can be cast into our framework as just being DPP-MAX where the maximization is done via a greedy selection rule.

Given that DPP-MAX is too greedy, it may be desirable to allow for uncertainty in the observations. Thus, we define DPP-SAMPLE which selects the batches via sampling subsets from DPPs, and show that the expected regret is smaller than that of DPP-MAX. To provide a fair comparison with an existing method, BUCB, we also derive regret bounds for B-EST [Theorem 3.2]. Finally, for all methods with known regret bounds, the key quantity is the information gain. In the appendix, we also provide a simpler proof of the information gain for the widely-used RBF kernel which also improves the bound from $\mathcal{O}((\log T)^{d+1})$ [26, 30] to $\mathcal{O}((\log T)^d)$. We conclude with experiments on synthetic and real-world robotics and hyper-parameter optimization for extreme multi-label classification tasks which demonstrate that our DPP-based methods, especially the sampling based ones are superior or competitive with the existing baselines.

**Related Work** One of the key tasks involved in black box optimization is of choosing actions that both explore the function and exploit our knowledge about likely high reward regions in the function's domain. This exploration-exploitation trade-off becomes especially important when the function is expensive to evaluate. This exploration-exploitation trade off naturally leads to modeling this problem in the multi-armed bandit paradigm [25], where the goal is to maximize cumulative reward by optimally balancing this trade-off. Srinivas *et al.* [30] analyzed the Gaussian Process Upper Confidence Bound (GP-UCB) algorithm, a simple and intuitive Bayesian method [3] to achieve the first sub-linear regret bounds for Gaussian process bandit optimization. These bounds however grow logarithmically in the size of the (finite) search space.

Recent work by Wang *et al.* [34] considered an intuitive MAP-estimate based strategy (EST) which involves estimating the maximum value of a function and choosing a point which has maximum probability of achieving this maximum value. They derive regret bounds for this strategy and show that the bounds are actually independent of the size of the search space. The problem setting for both UCB and EST is of optimizing a particular *acquisition function*. Other popular acquisition functions include expected improvement (EI), probability of improvement over a certain threshold (PI). Along with these, there is also work on Entropy search (ES) [13] and its variant, predictive entropy search (PES) [14] which instead aims at minimizing the uncertainty about the location of the optimum of the function. All the fore-mentioned methods, though, are inherently sequential in nature.

The BUCB and UCB-PE both depend on the crucial observation that the variance of the posterior distribution does not depend on the actual values of the function at the selected points. They exploit this fact by "hallucinating" the function values to be as predicted by the posterior mean. The BUCB algorithm chooses the batch by sequentially selecting the points with the maximum UCB score keeping the mean function the same and only updating the variance. The problem with this naive approach is that it is too "overconfident" of the observations which causes the confidence bounds on the function values to shrink very quickly as we go deeper into the batch. This is fixed by a careful initialization and expanding the confidence bounds which leads to regret bounds which are worse than that of UCB by some multiplicative factor (independent of T and B). The UCB-PE algorithm chooses the first point of the batch via the UCB score and then defines a "relevance region" and selects the remaining points from this region greedily to maximize the *information gain*, in order to focus on pure exploration (PE). This algorithm does not require any initialization like the BUCB and, in fact, achieves better regret bounds than the BUCB.

Both BUCB and UCB-PE, however, are too greedy in their selection of batches which may be really far from the optimal due to our "immediate overconfidence" of the values. Indeed this is the criticism of these two methods by a recently proposed BBO strategy PPES [27], which parallelizes predictive entropy search based methods and shows considerable improvements over the BUCB and UCB-PE methods. Another recently proposed method is the Local Penalization (LP) [11], which assumes that the function is Lipschitz continuous and tries to estimate the Lipschitz constant. Since assumptions of Lipschitz continuity naturally allow one to place bounds on how far the optimum of $f$ is from a certain location, they work to smoothly reduce the value of the acquisition function in a neighborhood of any point reflecting the belief about the distance of this point to the maxima. However, assumptions of Lipschitzness are too coarse-grained and it is unclear how their method to estimate the Lipschitz constant and modelling of local penalization affects the performance from a theoretical standpoint. Our algorithms, in constrast, are general and do not assume anything about the function other than it being drawn from a Gaussian Process.

## 2 Preliminaries

**Gaussian Process Bandit Optimization** We address the problem of finding, in the lowest possible number of iterations, the maximum $(m)$ of an unknown function $f : \mathcal{X} \to \mathbb{R}$ where $\mathcal{X} \subset \mathbb{R}^d$, i.e.,

$$m = f(x^*) = \max_{x \in \mathcal{X}} f(x).$$

We consider the domain to be discrete as it is well-known how to obtain regret bounds for continous, compact domains via suitable discretizations [30]. At each iteration $t$, we choose a batch $\{x_{t,b}\}_{1 \le b \le B}$ of $B$ points and then simultaneously observe the noisy values taken by $f$ at these points, $y_{t,b} = f(x_{t,b}) + \epsilon_{t,b}$, where $\epsilon_{t,k}$ is i.i.d. Gaussian noise $\mathcal{N}(0, \sigma^2)$. The function is assumed to be drawn from a Gaussian process (GP), i.e., $f \sim GP(0, k)$, where $k : \mathcal{X}^2 \to \mathbb{R}_+$ is the kernel function. Given the observations $\mathcal{D}_t = \{(x_\tau, y_\tau)_{\tau=1}^t\}$ up to time $t$, we obtain the posterior mean and covariance functions [24] via the kernel matrix $K_t = [k(x_i, x_j)]_{x_i, x_j \in \mathcal{D}_t}$ and $\mathbf{k_t}(x) = [k(x_i, x)]_{x_i \in \mathcal{D}_t} : \mu_t(x) = \mathbf{k_t}(x)^T (K_t + \sigma^2 I)^{-1} \mathbf{y_t}$ and $k_t(x, x') = k(x, x') - \mathbf{k_t}(x)^T (K_t + \sigma^2 I)^{-1} \mathbf{k_t}(x')$. The posterior variance is given by $\sigma_t^2(x) = k_t(x, x)$. Define the Upper Confidence Bound (UCB) $f^+$ and Lower Confidence Bound (LCB) $f^-$ as

$$f_t^+(x) = \mu_{t-1}(x) + \beta_t^{1/2} \sigma_{t-1}(x) \qquad f_t^-(x) = \mu_{t-1}(x) - \beta_t^{1/2} \sigma_{t-1}(x)$$

A crucial observation made in BUCB [9] and UCB-PE [8] is that the posterior covariance and variance functions do not depend on the actual function values at the set of points. The EST algorithm in [34] chooses at each timestep $t$,the point which has the maximum posterior probability of attaining the maximum value $m$, i.e., the $\arg\max_{x \in \mathcal{X}} \Pr(M_x | m, \mathcal{D}_t)$ where $M_x$ is the event that point $x$ achieves the maximum value. This turns out to be equal to $\arg\min_{x \in \mathcal{X}} \left[ (m - \mu_t(x)) / \sigma_t(x) \right]$. Note that this actually depends on the value of $m$ which, in most cases, is unknown. [34] get around this by using an approximation $\hat{m}$ which, under certain conditions specified in their paper, is an upper bound on $m$. They provide two ways to get the estimate $\hat{m}$, namely ESTa and ESTn. We refer the reader to [34] for details of the two estimates and refer to ESTa as EST.

Assuming that the horizon $T$ is unknown, a strategy has to be good at any iteration. Let $r_{t,b}$ denote the *simple regret*, the difference between the value of the maxima and the point queried $x_{t,k}$, i.e., $r_{t,b} = \max_{x \in \mathcal{X}} f(x) - f(x_{t,b})$. While, UCB-PE aims at minimizing a batched cumulative regret, in this paper we will focus on the standard full cumulative regret defined as $R_{TB} = \sum_{t=1}^T \sum_{b=1}^B r_{t,b}$. This models the case where all the queries in a batch should have low regret. The key quantity controlling the regret bounds of all known BO algorithms is the maximum mutual information that can be gained about $f$ from $T$ measurements : $\gamma_T = \max_{A \subseteq \mathcal{X}, |A| \le T} I(y_A, f_A) = \max_{A \subseteq \mathcal{X}, |A| \le T} \frac{1}{2} \log \det(I + \sigma^{-2} K_A)$, where $K_A$ is the (square) submatrix of $K$ formed by picking the row and column indices corresponding to the set $A$. The regret for both the UCB and the EST algorithms are presented in the following theorem which is a combination of Theorem 1 in [30] and Theorem 3.1 in [34].

**Theorem 2.1.** *Let $C = 2/\log(1 + \sigma^{-2})$ and fix $\delta > 0$. For UCB, choose $\beta_t = 2\log(|\mathcal{X}|t^2\pi^2/6\delta)$ and for EST, choose $\beta_t = (\min_{x \in \mathcal{X}} \frac{\hat{m} - \mu_{t-1}(x)}{\sigma_{t-1}(x)})^2$ and $\zeta_t = 2\log(\pi^2 t^2/\delta)$. With probability $1 - \delta$, the cumulative regret up to any time step $T$ can be bounded as*

$$R_T = \sum_{t=1}^T r_t \le \begin{cases} \sqrt{CT\beta_T\gamma_T} & \text{for UCB} \\ \sqrt{CT\gamma_T}(\beta_{t^*}^{1/2} + \zeta_T^{1/2}) & \text{for EST} \end{cases} \text{ where } t^* = \arg\max_t \beta_t.$$

**Determinantal Point Processes** Given a DPP kernel $K \in \mathbb{R}^{m \times m}$ of $m$ elements $\{1, \ldots, m\}$, the $k$-DPP distribution defined on $2^{\mathcal{Y}}$ is defined as picking $B$, a $k$-subset of $[m]$ with probability proportional

---

**Algorithm 1** GP-BUCB/B-EST Algorithm

---

**Input:** Decision set $\mathcal{X}$, GP prior $\mu_0, \sigma_0$, kernel function $k(\cdot, \cdot)$, feedback mapping $fb[\cdot]$

**for** $t = 1$ **to** TB **do**

   Choose $\beta_t^{1/2} = \begin{cases} C'\big[2\log(|\mathcal{X}|\pi^2 t^2/6)\delta\big] & \text{for BUCB} \\ C'\big[\min_{x\in\mathcal{X}}(\hat{m} - \mu_{fb[t]})/\sigma_{t-1}(x)\big] & \text{for B-EST} \end{cases}$

   Choose $x_t = \arg\max_{x\in\mathcal{X}}[\mu_{fb[t]}(x) + \beta_t^{1/2}\sigma_{t-1}(x)]$ and compute $\sigma_t(\cdot)$

   **if** $fb[t] < fb[t+1]$ **then**

      Obtain $y_{t'} = f(x_{t'}) + \epsilon_{t'}$ for $t' \in \{fb[t]+1, \ldots, fb[t+1]\}$ and compute $\mu_{fb[t+1]}(\cdot)$

   **end if**

**end for**

**return** $\arg\max\limits_{t=1\ldots TB} y_t$

---

to $\det(K_B)$. Formally,

$$\Pr(B) = \frac{\det(K_B)}{\sum_{|S|=k}\det(K_S)}$$

The problem of picking a set of size $k$ which maximizes the determinant and sampling a set according to the $k$-DPP distribution has received considerable attention [22, 7, 6, 10, 1, 17]. The maximization problem in general is NP-hard and furthermore, has a hardness of approximation result of $1/c^k$ for some $c > 1$. The best known approximation algorithm is by [22] with a factor of $1/e^k$, which almost matches the lower bound. Their algorithm however is a complicated and expensive convex program. A simple greedy algorithm on the other hand gives a $1/2^{k\log(k)}$-approximation. For sampling from $k$-DPPs, an exact sampling algorithm exists due to [10]. This, however, does not scale to large datasets. A recently proposed alternative is an MCMC based method by [1] which is much faster.

## 3 Main Results

In this section, we present our DPP-based algorithms. For a fair comparison of the various methods, we first prove the regret bounds of the EST version of BUCB, i.e., B-EST. We then show the equivalence between UCB-PE and UCB-DPP maximization along with showing regret bounds for the EST version of PE/DPP-MAX. We then present the DPP sampling (DPP-SAMPLE) based methods for UCB and EST and provide regret bounds. In Appendix 4, while borrowing ideas from [26], we provide a simpler proof with improved bounds on the maximum information gain for the RBF kernel.

### 3.1 The Batched-EST algorithm

The BUCB has a feedback mapping $fb$ which indicates that at any given time $t$ (just in this case we will mean a total of $TB$ timesteps), the iteration upto which the actual function values are available. In the batched setting, this is just $\lfloor(t-1)/B\rfloor B$. The BUCB and B-EST, its EST variant algorithms are presented in Algorithm 1. The algorithm mainly comes from the observation made in [34] that the point chosen by EST is the same as a variant of UCB. This is presented in the following lemma.

**Lemma 3.1.** *(Lemma 2.1 in [34]) At any timestep $t$, the point selected by EST is the same as the point selected by a variant of UCB with $\beta_t^{1/2} = \min_{x\in\mathcal{X}}(\hat{m} - \mu_{t-1}(x))/\sigma_{t-1}(x)$.*

This will be sufficient to get to B-EST as well by just running BUCB with the $\beta_t$ as defined in Lemma 3.1 and is also provided in Algorithm 1. In the algorithm, $C'$ is chosen to be $exp(2C)$, where $C$ is an upper bound on the maximum conditional mutual information $I(f(x); y_{fb[t]+1:t-1}|y_{1:fb[t]})$ (refer to [9] for details). The problem with naively using this algorithm is that the value of $C'$, and correspondingly the regret bounds, usually has at least linear growth in $B$. This is corrected in [9] by two-stage BUCB which first chooses an initial batch of size $T^{init}$ by greedily choosing points based on the (updated) posterior variances. The values are then obtained and the posterior GP is calculated which is used as the prior GP in Algorithm 1. The $C'$ value can then be chosen independent of $B$. We refer the reader to the Table 1 in [9] for values of $C'$ and $T^{init}$ for common kernels. Finally, the regret bounds of B-EST are presented in the next theorem.

**Theorem 3.2.** *Choose $\alpha_t = \big(\min_{x\in\mathcal{X}}\frac{\hat{m}-\mu_{fb[t]}(x)}{\sigma_{t-1}(x)}\big)^2$ and $\beta_t = (C')^2\alpha_t$, $B \geq 2, \delta > 0$ and the $C'$ and $T^{init}$ values are chosen according to Table 1 in [9]. At any timestep $T$, let $R_T$ be the cumulative regret of the two-stage initialized B-EST algorithm. Then*

$$Pr\{R_T \leq C'R_T^{seq} + 2\|f\|_\infty T^{init}, \forall T \geq 1\} \geq 1 - \delta$$

*Proof.* The proof is presented in Appendix 1. □

**Algorithm 2** GP-(UCB/EST)-DPP-(MAX/SAMPLE) Algorithm

---

**Input:** Decision set $\mathcal{X}$, GP prior $\mu_0, \sigma_0$, kernel function $k(\cdot, \cdot)$
**for** $t = 1$ **to** T **do**
    Compute $\mu_{t-1}$ and $\sigma_{t-1}$ according to Bayesian inference.
    Choose $\beta_t^{1/2} = \begin{cases} \left[2\log(|\mathcal{X}|\pi^2 t^2/6)\delta\right] & \text{for UCB} \\ \left[\min_{x \in \mathcal{X}}(\hat{m} - \mu_{fb[t]})/\sigma_{t-1}(x)\right] & \text{for EST} \end{cases}$
    $x_{t,1} \leftarrow \arg\max_{x \in \mathcal{X}} \mu_{t-1}(x) + \sqrt{\beta_t}\sigma_{t-1}(x)$
    Compute $\mathcal{R}_t^+$ and construct the DPP kernel $K_{t,1}$
    $\{x_{t,b}\}_{b=2}^B \leftarrow \begin{cases} \mathsf{kDPPMaxGreedy}(K_{t,1}, B-1) & \text{for DPP-MAX} \\ \mathsf{kDPPSample}(K_{t,1}, B-1) & \text{for DPP-SAMPLE} \end{cases}$
    Obtain $y_{t,b} = f(x_{t,b}) + \epsilon_{t,b}$ for $b = 1, \ldots, B$
**end for**

---

## 3.2 Equivalence of Pure Exploration (PE) and DPP Maximization

We now present the equivalence between the Pure Exploration and a procedure which involves DPP maximization based on the Greedy algorithm. For the next two sections, by an iteration, we mean all $B$ points selected in that iteration and thus, $\mu_{t-1}$ and $k_{t-1}$ are computed using $(t-1)B$ observations that are available to us. We first describe a generic framework for BBO inspired by UCB-PE : At any iteration, the first point is chosen by selecting the one which maximizes UCB or EST which can be seen as a variant of UCB as per Lemma 3.1. A relevance region $\mathcal{R}_t^+$ is defined which contains $\arg\max_{x \in \mathcal{X}} f_{t+1}^+(x)$ with high probability. Let $y_t^\bullet = f_t^-(x_t^\bullet)$, where $x_t^\bullet = \arg\max_{x \in \mathcal{X}} f_t^-(x)$. The relevance region is formally defined as $\mathcal{R}_t^+ = \{x \in \mathcal{X} | \mu_{t-1} + 2\sqrt{\beta_{t+1}}\sigma_{t-1}(x) \geq y_t^\bullet\}$. The intuition for considering this region is that using $\mathcal{R}_t^+$ guarantees that the queries at iteration $t$ will leave an impact on the future choices at iteration $t+1$. The next $B-1$ points for the batch are then chosen from $\mathcal{R}_t^+$, according to some rule. In the special case of UCB-PE, the $B-1$ points are selected greedily from $\mathcal{R}_t^+$ by maximizing the (updated) posterior variance, while keeping the mean function the same. Now, at the $t^{th}$ iteration, consider the posterior kernel function after $x_{t,1}$ has been chosen (say $k_{t,1}$) and consider the kernel matrix $K_{t,1} = I + \sigma^{-2}[k_{t,1}(p_i, p_j)]_{i,j}$ over the points $p_i \in \mathcal{R}_t^+$. We will consider this as our DPP kernel at iteration $t$. Two possible ways of choosing $B-1$ points via this DPP kernel is to either choose the subset of size $B-1$ of maximum determinant (DPP-MAX) or sample a set from a $(B-1)$-DPP using this kernel (DPP-SAMPLE). In this subsection, we focus on the maximization problem. The proof of the regret bounds of UCB-PE go through a few steps but in one of the intermediate steps (Lemma 5 of [8]), it is shown that the sum of regrets over a batch at an iteration $t$ is upper bounded as

$$\sum_{b=1}^B r_{t,b} \leq \sum_{b=1}^B (\sigma_{t,b}(x_{t,b}))^2 \leq \sum_{b=1}^B C_2\sigma^2 \log(1 + \sigma^{-2}\sigma_{t,b}(x_{t,b})) = C_2\sigma^2 \log\left[\prod_{b=1}^B (1 + \sigma^{-2}\sigma_{t,b}(x_{t,b}))\right]$$

where $C_2 = \sigma^{-2}/\log(1 + \sigma^{-2})$. From the final log-product term, it can be seen (from Schur's determinant identity [5] and the definition of $\sigma_{t,b}(x_{t,b})$) that the product of the last $B-1$ terms is exactly the $B-1$ principal minor of $K_{t,1}$ formed by the indices corresponding to $S = \{x_{t,b}\}_{b=2}^B$. Thus, it is straightforward to see that the UCB-PE algorithm is really just $(B-1)$-DPP maximization via the greedy algorithm. This connection will also be useful in the next subsection for DPP-SAMPLE. Thus, $\sum_{b=1}^B r_{t,b} \leq C_2\sigma^2\left[\log(1 + \sigma^{-2}\sigma_{t,1}(x_{t,1})) + \log\det((K_{t,1})_S)\right]$. Finally, for EST-PE, the proof proceeds like in the B-EST case by realising that EST is just UCB with an adaptive $\beta_t$. The final algorithm (along with its sampling counterpart; details in the next subsection) is presented in Algorithm 2. The procedure $\mathsf{kDPPMaxGreedy}(K, k)$ picks a principal submatrix of $K$ of size $k$ by the greedy algorithm. Finally, we have the theorem for the regret bounds for (UCB/EST)-DPP-MAX.

**Theorem 3.3.** *At iteration $t$, let $\beta_t = 2\log(|\mathcal{X}|\pi^2 t^2/6\delta)$ for UCB, $\beta_t = (\min \frac{\hat{m} - \mu_{t-1}(x)}{\sigma_{t-1}(x)})^2$ and $\zeta_t = 2\log(\pi^2 t^2/3\delta)$ for EST, $C_1 = 36/\log(1 + \sigma^{-2})$ and fix $\delta > 0$, then, with probability $\geq 1 - \delta$ the full cumulative regret $R_{TB}$ incurred by UCB-DPP-MAX is $R_{TB} \leq \sqrt{C_1 TB\beta_T\gamma_{TB}}\}$ and that for EST-DPP-MAX is $R_{TB} \leq \sqrt{C_1 TB\gamma_{TB}}(\beta_{t*}^{1/2} + \zeta_T^{1/2})$.*

*Proof.* The proof is provided in Appendix 2. It should be noted that the term inside the logarithm in $\zeta_t$ has been multiplied by 2 as compared to the sequential EST, which has a union bound over just one point, $x_t$. This happens because we will need a union bound over not just $x_{t,b}$ but also $x_t^\bullet$. $\square$

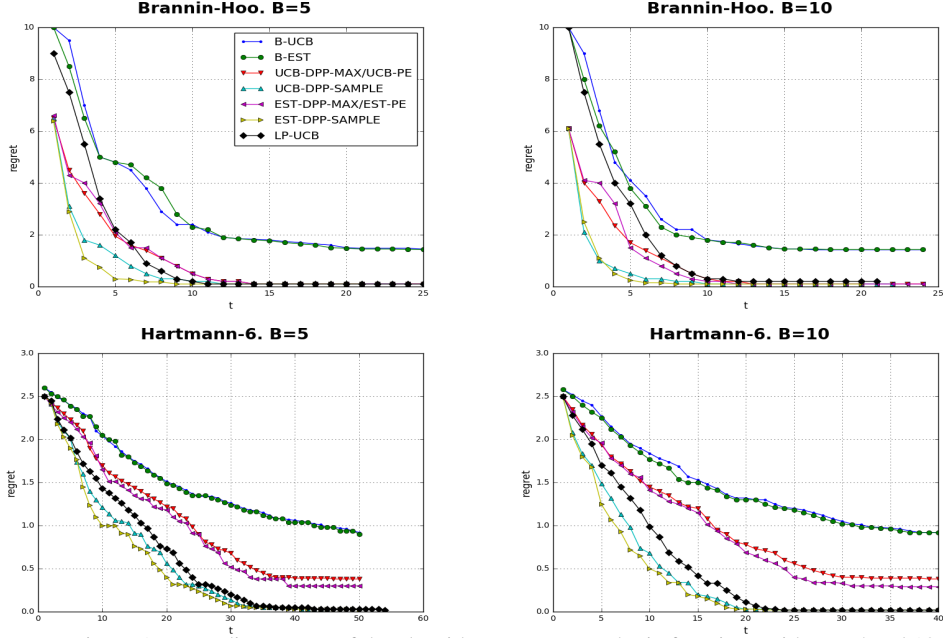

Figure 1: Immediate regret of the algorithms on two synthetic functions with B = 5 and 10

### 3.3 Batch Bayesian Optimization via DPP Sampling

In the previous subsection, we looked at the regret bounds achieved by DPP maximization. One natural question to ask is whether the other subset selection method via DPPs, namely DPP sampling, gives us equivalent or better regret bounds. Note that in this case, the regret would have to be defined as expected regret. The reason to believe this is well-founded as indeed sampling from $k$-DPPs results in better results, in both theory and practice, for low-rank matrix approximation [10] and exemplar-selection for Nystrom methods [19]. Keeping in line with the framework described in the previous subsection, the subset to be selected has to be of size $B - 1$ and the kernel should be $K_{t,1}$ at any iteration $t$. Instead of maximizing, we can choose to sample from a $(B - 1)$-DPP. The algorithm is described in Algorithm 2. The kDPPSample(K, k) procedure denotes sampling a set from the $k$-DPP distribution with kernel $K$. The question then to ask is what is the expected regret of this procedure. In this subsection, we show that the expected regret bounds of DPP-SAMPLE are less than the regret bounds of DPP-MAX and give a quantitative bound on this regret based on entropy of DPPs. By entropy of a $k$-DPP with kernel $K$, $H(k - \text{DPP}(K))$, we simply mean the standard definition of entropy for a discrete distribution. Note that the entropy is always non-negative in this case. Please see Appendix 3 for details. For brevity, since we always choose $B - 1$ elements from the DPP, we denote $H(DPP(K))$ to be the entropy of $(B - 1)$-DPP for kernel $K$.

**Theorem 3.4.** *The regret bounds of DPP-SAMPLE are less than that of DPP-MAX. Furthermore, at iteration $t$, let $\beta_t = 2\log(|\mathcal{X}|\pi^2 t^2/6\delta)$ for UCB, $\beta_t = (\min \frac{\hat{m}-\mu_{t-1}(x)}{\sigma_{t-1}(x)})^2$ and $\zeta_t = 2\log(\pi^2 t^2/3\delta)$ for EST, $C_1 = 36/\log(1 + \sigma^{-2})$ and fix $\delta > 0$, then the expected full cumulative regret of UCB-DPP-SAMPLE satisfies*

$$R_{TB}^2 \leq 2TBC_1\beta_T\left[\gamma_{TB} - \sum_{t=1}^{T} H(DPP(K_{t,1})) + B\log(|\mathcal{X}|)\right]$$

*and that for EST-DPP-SAMPLE satisfies*

$$R_{TB}^2 \leq 2TBC_1(\beta_t^{1/2} + \zeta_t^{1/2})^2\left[\gamma_{TB} - \sum_{t=1}^{T} H(DPP(K_{t,1})) + B\log(|\mathcal{X}|)\right]$$

*Proof.* The proof is provided in Appendix 3. □

Note that the regret bounds for both DPP-MAX and DPP-SAMPLE are better than BUCB/B-EST due to the latter having both an additional factor of $B$ in the $\log$ term and a regret multiplier constant $C'$. In fact, for the RBF kernel, $C'$ grows like $e^{d^d}$ which is quite large for even moderate values of $d$.

## 4 Experiments

In this section, we study the performance of the DPP-based algorithms, especially DPP-SAMPLE against some existing baselines. In particular, the methods we consider are BUCB [9], B-EST,

UCB-PE/UCB-DPP-MAX [8], EST-PE/EST-DPP-MAX, UCB-DPP-SAMPLE, EST-DPP-SAMPLE and UCB with local penalization (LP-UCB) [11]. We used the publicly available code for BUCB and PE[1]. The code was modified to include the code for the EST counterparts using code for EST [2]. For LP-UCB, we use the publicly available GPyOpt codebase [3] and implemented the MCMC algorithm by [1] for $k$-DPP sampling with $\epsilon = 0.01$ as the variation distance error. We were unable to compare against PPES as the code was not publicly available. Furthermore, as shown in the experiments in [27], PPES is very slow and does not scale beyond batch sizes of 4-5. Since UCB-PE almost always performs better than the simulation matching algorithm of [4] in all experiments that we could find in previous papers [27, 8], we forego a comparison against simulation matching as well to avoid clutter in the graphs. The performance is measured after $t$ batch evaluations using *immediate regret*, $r_t = |f(\widetilde{x}_t) - f(x^*)|$, where $x^*$ is a known optimizer of $f$ and $\widetilde{x}_t$ is the recommendation of an algorithm after $t$ batch evaluations. We perform 50 experiments for each objective function and report the median of the immediate regret obtained for each algorithm. To maintain consistency, the first point of all methods is chosen to be the same (random). The mean function of the prior GP was the zero function while the kernel function was the squared-exponential kernel of the form $k(x, y) = \gamma^2 \exp[-0.5 \sum_d (x_d - y_d^2)/l_d^2]$. The hyper-parameter $\lambda$ was picked from a broad Gaussian hyperprior and the the other hyper-parameters were chosen from uninformative Gamma priors.

Our first set of experiments is on a set of synthetic benchmark objective functions including Branin-Hoo [20], a mixture of cosines [2] and the Hartmann-6 function [20]. We choose batches of size 5 and 10. Due to lack of space, the results for mixture of cosines are provided in Appendix 5 while the results of the other two are shown in Figure 1. The results suggest that the DPP-SAMPLE based methods perform superior to the other methods. They do much better than their DPP-MAX and Batched counterparts. The trends displayed with regards to LP are more interesting. For the Branin-Hoo, LP-UCB starts out worse than the DPP based algorithms but takes over DPP-MAX relatively quickly and approaches the performance of DPP-SAMPLE when the batch size is 5. When the batch size is 10, the performance of LP-UCB does not improve much but both DPP-MAX and DPP-SAMPLE perform better. For Hartmann, LP-UCB outperforms both DPP-MAX algorithms by a considerable margin. The DPP-SAMPLE based methods perform better than LP-UCB. The gap, however, is more for the batch size of 10. Again, the performance of LP-UCB changes much lesser compared to the performance gain of the DPP-based algorithms. This is likely because the batches chosen by the DPP-based methods are more "globally diverse" for larger batch sizes. The superior performance of the sampling based methods can be attributed to allowing for uncertainty in the observations by sampling as opposed to greedily emphasizing on maximizing information gain.

We now consider maximization of real-world objective functions. The first function we consider, `robot`, returns the walking speed of a bipedal robot [35]. The function's input parameters, which live in $[0, 1]^8$, are the robot's controller. We add Gaussian noise with $\sigma = 0.1$ to the noiseless function. The second function, `Abalone`[4] is a test function used in [8]. The challenge of the dataset is to predict the age of a species of sea snails from physical measurements. Similar to [8], we will use it as a maximization problem. Our final experiment is on hyper-parameter tuning for extreme multi-label learning. In extreme classification, one needs to deal with multi-class and multi-label problems involving a very large number of categories. Due to the prohibitively large number of categories, running traditional machine learning algorithms is not feasible. A recent popular approach for extreme classification is the FastXML algorithm [23]. The main advantage of FastXML is that it maintains high accuracy while training in a fraction of the time compared to the previous state-of-the-art. The FastXML algorithm has 5 parameters and the performance depends on these hyper-parameters, to a reasonable amount. Our task is to perform hyper-parameter optimization on these 5 hyper-parameters with the aim to maximize the Precision@k for $k = 1$, which is the metric used in [23] to evaluate the performance of FastXML compared to other algorithms as well. While the authors of [23] run extensive tests on a variety of datasets, we focus on two small datasets : Bibtex [15] and Delicious[32]. As before, we use batch sizes of 5 and 10. The results for Abalone and the FastXML experiment on Delicious are provided in the appendix. The results for Prec@1 for FastXML on the Bibtex dataset

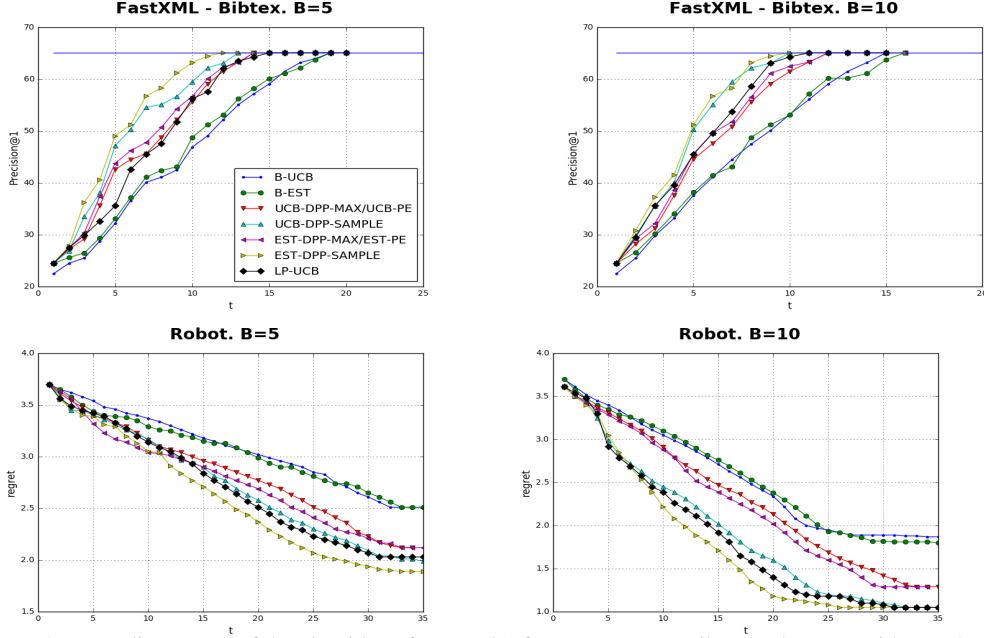

Figure 2: Immediate regret of the algorithms for Prec@1 for FastXML on Bibtex and `Robot` with B = 5 and 10

and for the `robot` experiment are provided in Figure 2. The blue horizontal line for the FastXML results indicates the maximum Prec@k value found using grid search.

The results for `robot` indicate that while DPP-MAX does better than their Batched counterparts, the difference in the performance between DPP-MAX and DPP-SAMPLE is much less pronounced for a small batch size of 5 but is considerable for batch sizes of 10. This is in line with our intuition about sampling being more beneficial for larger batch sizes. The performance of LP-UCB is quite close and slightly better than UCB-DPP-SAMPLE. This might be because the underlying function is well-behaved (Lipschitz continuous) and thus, the estimate for the Lipschitz constant might be better which helps them get better results. This improvement is more pronounced for batch size of 10 as well. For Abalone (see Appendix 5), LP does better than DPP-MAX but there is a reasonable gap between DPP-SAMPLE and LP which is more pronounced for $B = 10$.

The results for Prec@1 for the Bibtex dataset for FastXML are more interesting. Both DPP based methods are much better than their Batched counterparts. For $B = 5$, DPP-SAMPLE is only slightly better than DPP-MAX. LP-UCB starts out worse than DPP-MAX but starts doing comparable to DPP-MAX after a few iterations. For $B = 10$, there is not a large improvement in the gap between DPP-MAX and DPP-SAMPLE. LP-UCB however, quickly takes over UCB-DPP-MAX and comes quite close to the performance of DPP-SAMPLE after a few iterations. For the Delicious dataset (see Appendix 5), we see a similar trend of the improvement of sampling to be larger for larger batch sizes. LP-UCB displays an interesting trend in this experiment by doing much better than UCB-DPP-MAX for $B = 5$ and is in fact quite close to the performance of DPP-SAMPLE. However, for $B = 10$, its performance is much closer to UCB-DPP-MAX. DPP-SAMPLE loses out to LP-UCB only on the `robot` dataset and does better for all the other datasets. Furthermore, this improvement seems more pronounced for larger batch sizes. We leave experiments with other kernels and a more thorough experimental evaluation with respect to batch sizes for future work.

## 5 Conclusion

We have proposed a new method for batched Gaussian Process bandit (batch Bayesian) optimization based on DPPs which are desirable in this case as they promote diversity in batches. The DPP kernel is automatically figured out on the fly which allows us to show regret bounds for DPP maximization and sampling based methods for this problem. We show that this framework exactly recovers a popular algorithm for BBO, namely the UCB-PE when we consider DPP maximization using the greedy algorithm. We showed that the regret for the sampling based method is always less than the maximization based method. We also derived their EST counterparts and also provided a simpler proof of the information gain for RBF kernels which leads to a slight improvement in the best bound known. Our experiments on a variety of synthetic and real-world tasks validate our theoretical claims that sampling performs better than maximization and other methods.

## Footnotes

[1]http://econtal.perso.math.cnrs.fr/software/

[2]https://github.com/zi-w/EST

[3]http://sheffieldml.github.io/GPyOpt/

[4]The Abalone dataset is provided by the UCI Machine Learning Repository at http://archive.ics.uci.edu/ml/datasets/Abalone

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
