[Supplementary Material · appendix.pdf]

# Supplementary Material for "Batched Gaussian Process Bandit Optimization via Determinantal Point Processes"

## 1 The Batched-EST Algorithm

The proofs for B-EST are relatively straightforward which follow from combining the proofs of [2] and [5]. We provide them here for completeness. We first need a series of supporting lemmas which are variants of the lemmas of UCB for EST. These require different bounds than the ones for BUCB.

**Lemma 1.1.** *(Lemma 3.2 in [5]) Pick $\delta \in (0,1)$ and set $\zeta_t = 2\log(\pi^2 t^2 / 6\delta)$. Then, for an arbitrary sequence of actions $x_1, x_2, \ldots \in \mathcal{X}$,*

$$\Pr[|f(x_t) - \mu_{t-1}(x_t)| \leq \zeta_t^{1/2} \sigma_{t-1}(x_t)] \geq 1 - \delta, \text{ for all } t \in [1, T].$$

The GP-UCB/EST decision rule is,

$$x_t = \arg\max_{x \in \mathcal{X}} \left[ \mu_{t-1}(x) + \alpha_t^{1/2} \sigma_{t-1}(x) \right]$$

For EST, $\alpha_t = \left( \frac{m - \mu_{t-1}(x)}{\sigma_{t-1}(x)} \right)^2$. Implicit in this definition of the decision rule is the corresponding confidence interval for each $x \in \mathcal{X}$,

$$C_t^{seq}(x) \equiv \left[ \mu_{t-1}(x) - \alpha_t^{1/2} \sigma_{t-1}(x), \mu_{t-1}(x) + \alpha_t^{1/2} \sigma_{t-1}(x) \right],$$

where this confidence interval's upper confidence bound is the value of the argument of the decision rule. Furthermore, the width of any confidence interval is the difference between the uppermost and the lowermost limits, here $w = 2\alpha_t^{1/2} \sigma_{t-1}(x)$. In the case of BUCB/B-EST, the batched confidence rules are of the form,

$$C_t^{batch}(x) \equiv \left[ \mu_{fb[t]}(x) - \beta_t^{1/2} \sigma_{t-1}(x), \mu_{fb[t]}(x) + \beta_t^{1/2} \sigma_{t-1}(x) \right]$$

**Lemma 1.2.** *(Similar to Lemma 12 in [2]) If $f(x_t) \in C_t^{batch}(x_t) \forall t \geq 1$ and given that actions are selected using EST, it holds that,*

$$R_T \leq \sqrt{TC_1 \gamma_T} (\zeta_T^{1/2} + \alpha_{t^*}^{1/2})$$

*Proof.* The proof of Lemma 12 in [2] just uses the fact that the sequential regret bounds of UCB are $2\beta_t^{1/2} \sigma_t(x_t)$. We follow their same proof but use the EST sequential bounds to get the desired result. $\square$

*Proof.* (of Theorem 3.2 in the main paper) The proof is similar to Theorem 5 in [2]. The sum of the regrets over the T timesteps is split over the first $T^{init}$ timesteps and the remaining timesteps. The former term $\sum_{t=1}^{T^{init}} m - f(x_t) \leq 2T^{init} \|f\|_\infty$. The latter term is treated as simple BUCB and from Lemma 1.2, we get $R_{T^{init}+1:T} \leq \sqrt{(T - T^{init})C_1 \gamma_{(T-T^{init})}} (\zeta_T^{1/2} + \alpha_{t^*}^{1/2}) \leq \sqrt{TC_1 \gamma_T} (\zeta_T^{1/2} + \alpha_{t^*}^{1/2})$ as $\gamma_T$ is a non-decreasing function. Combining the two terms gives us the desired result. $\square$

## 2 Batch Bayesian Optimization via DPP-Maximization

In this section, we present the proof of the regret bounds for BBO via DPP-MAX. Since the GP-UCB-DPP-MAX is the same as GP-UCB-PE, we focus on GP-EST-DPP-MAX. We first restate the EST part of Theorem 3.3. Firstly, none of our proofs will depend on the order in which the batch was constructed but for sake of clarity of exposition, whenever needed, we can consider any arbitrary ordering of the $B - 1$ points chosen by maximizing a $(B - 1)$-DPP or sampling from it.

**Theorem 2.1.** *At iteration $t$, fix $\delta > 0$ and let $\beta_t = \left[\min_{x \in \mathcal{X}} \frac{m - \mu_{t-1}(x)}{\sigma_{t-1}(x)}\right]$, $\zeta_t = 2\log(\pi^2 t^2 / 3\delta)$ and $C_1 = 36/\log(1 + \sigma^{-2})$. Then, with probability $\geq 1 - \delta$, the full cumulative regret incurred by EST-DPP-MAX is $R_t \leq \sqrt{C_1 TB\gamma_{TB}}(\beta_{t^*}^{1/2} + \zeta_{TB}^{1/2})$.*

Notice that the logarithm term for $\zeta_t$ in the above theorem is twice that of the one in Lemma 1.1. This happens by considering the same proof as that for Lemma 1.1 but taking a union bound over $x_t^\bullet$ along with $x_t$. We first prove some required lemmas.

**Lemma 2.2.** *The deviation of the first point, selected by either UCB or EST, is bounded by the deviation of any point selected by the DPP-MAX or DPP-SAMPLE in the previous iteration with high probability, i.e.,*

$$\forall t < T, \forall 2 \leq b \geq B, \quad \sigma_{t,1}(x_{t+1,1}) \leq \sigma_{t-1,b}(x_{t,b})$$

*Proof.* The proof does not depend on the actual policy (UCB/EST) used for the first point of the batch or whether it was DPP-MAX or DPP-SAMPLE (consider an arbitary ordering of points chosen in either case). By the definition of $x_{t+1,1}$, we have $f_{t+1}^+(x_{t+1,1}) \geq f_{t+1}^+(x_t^\bullet)$. Also, from Lemma 1.1, we have $f_{t+1}^+(x_t^\bullet) \geq f_t^-(x_t^\bullet)$. This is different than Lemma 2 in [1] as it now only holds for $x_t^\bullet$ rather than all $x \in \mathcal{X}$. Thus, with high probability, $f_{t+1}^+(x_{t+1,1}) \geq y_t^\bullet$ and thus, $x_{t+1,1} \in \mathcal{R}_t^+$. Now, from the definition of $x_{t,b}$, we have $\sigma_{t-1,b}(x_{t+1,1}) \leq \sigma_{t-1,b}(x_{t,b})$ w.h.p. Using the "Information never hurts" principle [3], we know that the entropy of $f(x)$ for all locations $x$ can only decrease after observing a point $x_{t,k}$. For GPs, the entropy is also a non-decreasing function of the variance and thus, we have, $\sigma_{t,1}(x_{t+1,1}) \leq \sigma_{t-1,b}(x_{t,b})$ and we are done. $\square$

**Lemma 2.3.** *(Lemma 3 in [1]) The sum of deviations of the points selected by the UCB/EST policy is bounded by the sum of deviations over all the selected points divided by B. Formally, with high probability,*

$$\sum_{t=1}^{T} \sigma_{t-1,1}(x_{t,1}) \leq \frac{1}{B}\sum_{t=1}^{T}\sum_{b=1}^{B} \sigma_{t-1,b}(x_{t,b}).$$

*Proof.* The proof is the same as that of Lemma 3 in [1] but we provide it here for completeness. Using Lemma 2.2 and the definitions of $x_{t,b}$, we have $\sigma_{t,1}(x_{t+1,1}) \leq \sigma_{t-1,b}$ for all $b \geq 2$. Summing over all $b$, we get for all $t \geq 1$, $\sigma_{t-1,1}(x_{t,1}) + (B-1)\sigma_{t,1}(x_{t+1,1}) \leq \sum_{b=1}^{B}\sigma_{t-1,b}(x_{t,b})$. Now, summing over $t$, we get the desired result. $\square$

**Lemma 2.4.** *(Lemma 4 in [1]) The sum of the variances of the selected points are bounded by a constant factor times $\gamma_{TB}$, i.e., $\exists C_1' \in \mathbb{R}$, $\sum_{t=1}^{T}\sum_{b=1}^{B}(\sigma_{t-1,b}(x_{t-1,b}))^2 \leq C_1'\gamma_{TB}$. Here $C_1' = \frac{2}{\log(1+\sigma^{-2})}$.*

We finally prove Theorem 2.1.

*Proof.* (of Theorem 2.1) Clearly, the proof of Lemma 1.1 holds even for the last $B - 1$ points selected in a batch. However, $t^*$ only goes over the the $T$ iterations rather than $TB$ evaluations. Thus, the cumulative regret is of the form

$$
\begin{aligned}
R_{TB} &= \sum_{t=1}^{T}\sum_{b=1}^{B} r_{t,b} \\
&\leq \sum_{t=1}^{T}\sum_{b=1}^{B} (\beta_{t^*}^{1/2} + \zeta_{TB}^{1/2})\sigma_{t-1,b}(x_{t,b}) \\
&\leq (\beta_{t^*}^{1/2} + \zeta_{TB}^{1/2})\sqrt{TB\sum_{t=1}^{T}\sum_{b=1}^{B}(\sigma_{t-1,b}(x_{t,b}))^2} \quad \text{by Cauchy-Schwarz} \\
&\leq \sqrt{TBC_1'\gamma_{TB}}(\beta_{t^*}^{1/2} + \zeta_{TB}^{1/2}) \quad \text{by Lemma 2.4} \\
&\leq \sqrt{TBC_1\gamma_{TB}}(\beta_{t^*}^{1/2} + \zeta_{TB}^{1/2}) \quad \text{since } C_1' \leq C_1
\end{aligned}
$$

$\square$

# 3 Batch Bayesian Optimization via DPP sampling

In this section, we prove the expected regret bounds obtained by DPP-SAMPLE (Theorem 3.4 of the main paper)

**Lemma 3.1.** *For the points chosen by (UCB/EST)-DPP-SAMPLE, the inequality,*

$$\sum_{t=1}^{T}(\sigma_{t-1,1}(x_{t-1,1}))^2 \leq \frac{1}{B-1}\sum_{t=1}^{T}\sum_{b=2}^{B}(\sigma_{t-1,b}(x_{t-1,b}))^2$$

*holds with high probability.*

*Proof.* Clearly, Lemma 2.3 holds in this case as well. Furthermore, it is easy to see that the inequality obtained by replacing every term in every summation by its square in Lemma 2.3 is also true by a similar proof. Thus, we have

$$\sum_{t=1}^{T}(\sigma_{t-1,1}(x_{t,1}))^2 \leq \frac{1}{B}\sum_{t=1}^{T}\sum_{b=1}^{B}(\sigma_{t-1,b}(x_{t,b}))^2$$

$$\implies (1-\frac{1}{B})\sum_{t=1}^{T}(\sigma_{t-1,1}(x_{t,1}))^2 \leq \frac{1}{B}\sum_{t=1}^{T}\sum_{b=2}^{B}(\sigma_{t-1,b}(x_{t,b}))^2$$

$$\implies \sum_{t=1}^{T}(\sigma_{t-1,1}(x_{t,1}))^2 \leq \frac{1}{B-1}\sum_{t=1}^{T}\sum_{b=2}^{B}(\sigma_{t-1,b}(x_{t,b}))^2$$

Hence, we are done. $\qquad\square$

Define $\eta_t^{1/2} = \begin{cases} 2\beta_t^{1/2} & \text{for UCB} \\ (\beta_{t^*}^{1/2} + \zeta_t^{1/2}) & \text{for EST} \end{cases}$.

*Proof.* (of Theorem 3.4 in the main paper) The expectation here is taken over the last $B-1$ points in each iteration being drawn from the $(B-1)$-DPP with the posterior kernel at the $t^{th}$ iteration. Using linearity of expectation, we get

$$\left(\mathbb{E}\big[\sum_{t=1}^{T}\sum_{b=1}^{B}r_{t,b}\big]\right)^2 = \left(\sum_{t=1}^{T}\mathbb{E}\big[\sum_{b=1}^{B}r_{t,b}\big]\right)^2$$

$$\leq \left(\sum_{t=1}^{T}\eta_{tB}^{1/2}\mathbb{E}\big[\sum_{b=1}^{B}\sigma_{t-1,b}(x_{t,b})\big]\right)^2$$

$$\leq \eta_{TB}TB\sum_{t=1}^{T}\mathbb{E}\big[\sum_{b=1}^{B}(\sigma_{t-1,b}(x_{t,b}))^2\big] \qquad \text{by Cauchy-Schwarz}$$

$$\leq \eta_{TB}\frac{TB^2}{B-1}\sum_{t=1}^{T}\mathbb{E}\big[\sum_{b=2}^{B}(\sigma_{t-1,b}(x_{t,b}))^2\big] \qquad \text{by Lemma 3.1}$$

It is easy to see by Schur's identity and the definition of $\sigma_{t-1,b}$ that the term inside the expectation is just $\log\det((K_{t,1})_S)$, where $S$ is the set of $B-1$ points chosen in the $t^{th}$ iteration by DPP sampling with kernel $K_{t,1}$. Let $L = B-1$. Thus,

$$\left(\mathbb{E}\big[\sum_{t=1}^{T}\sum_{b=1}^{B}r_{t,b}\big]\right)^2 \leq \eta_{TB}\frac{TB^2}{B-1}\sum_{t=1}^{T}\mathbb{E}_{S\sim(L-DPP(K_{t,1}))}\big[\log\det((K_{t,1})_S)\big]$$

Firstly, since the expectation is less than the maximum and $B/(B-1) \leq 2$, we get that the expected regret has the same bound as the regret bounds for DPP-MAX. This bound is however, loose. We get a bound below which may be worse but that is due to a loose analysis on our part and we can just choose the minimum of below and the DPP-MAX regret bounds. Expanding the expectation, we get

$$\mathbb{E}_{S \sim (L-DPP(K_{t,1}))}\left[\log \det((K_{t,1})_S)\right] = \sum_{|S|=L} \frac{\det((K_{t,1})_S) \log(\det((K_{t,1})_S))}{\sum\limits_{|S|=L} \det((K_{t,1})_S)}$$

$$= \sum_{|S|=L} \frac{\det((K_{t,1})_S) \log\left(\frac{\det((K_{t,1})_S)}{\sum\limits_{|S|=L} \det((K_{t,1})_S)}\right)}{\sum\limits_{|S|=L} \det((K_{t,1})_S)} + \frac{\det((K_{t,1})_S)) \log\left(\sum\limits_{|S|=L} \det((K_{t,1})_S)\right)}{\sum\limits_{|S|=L} \det((K_{t,1})_S)}$$

$$= -H(\text{L-DPP}(K_{t,1})) + \log\left(\sum_{|S|=L} \det((K_{t,1})_S)\right)$$

$$\leq -H(\text{L-DPP}(K_{t,1})) + \log(|\mathcal{X}|^L \max \det((K_{t,1})_S))$$

$$\leq -H(\text{L-DPP}(K_{t,1})) + L\log(|\mathcal{X}|) + \log(\max \det((K_{t,1})_S))$$

Plugging this into the summation and observing that the summation over last term is less than $C_1' \gamma_{TB}$, we get the desired result. □

# 4   Bounds on Information Gain for RBF kernels

**Theorem 4.1.** *The maximum information gain for the RBF kernel after $S$ timesteps is $\mathcal{O}\left((\log|S|)^d\right)$*

*Proof.* Let $x_S$ be the vector of points from subset $S$, that is, $x_S = (x)_{x \in S}$, and the noisy evaluations of a function $f$ at these points be denoted by a vector $y_S = f_S + \epsilon_S$, where $f_S = (f(x))_{x \in S}$ and $\epsilon_S \sim N(0, \sigma^2 I)$. In Bayesian experimental design, the informativeness or the information gain of $S$ is given by the mutual information between $f$ and these observations $I(y_S; f) = H(y_S) - H(y_S \mid f)$. When $f$ is modeled by a Gaussian process, it is specified by the mean function $\mu(x) = \mathbb{E}f(x)$ and the covariance or kernel function $k(x, x') = \mathbb{E}(f(x) - \mu(x))(f(x') - \mu(x'))$. In this case,

$$I(y_S; f) = I(y_S; f_S) = \frac{1}{2} \log \det(I + \sigma^{-2} K_S),$$

where $K_S = (k(x, x'))_{x,x' \in S}$. It is easy to see that

$$\log \det(I + \sigma^{-2} K_S) = \sum_{t=1}^{|S|} \log\left(1 + \sigma^{-2} \lambda_t(K_S)\right)$$

Seeger *et al.* [4] showed that for a Gaussian RBF kernel in $d$ dimensions, $\lambda_t(K) \leq cB^{t^{1/d}}$, with $B < 1$. Let $T = \left(\log_{1/B} |S|\right)^d \ll |S|$. Then for $t > T$, we have $\lambda_t \leq c/T$, and for $t \leq T$, we have $\lambda_t \leq c$. Therefore,

$$\log \det(I + \sigma^{-2} K_S) = \sum_{t=1}^{|S|} \log\left(1 + \sigma^{-2} \lambda_t(K_S)\right)$$

$$= \sum_{t \leq T} \log\left(1 + \sigma^{-2} \lambda_t(K_S)\right) + \sum_{t > T} \log\left(1 + \sigma^{-2} \lambda_t(K_S)\right)$$

$$\leq T \log\left(1 + \sigma^{-2} c\right) + \log\left(1 + \frac{c\sigma^{-2}}{|S|}\right)^T$$

$$= O(T)$$

Thus, the maximum information gain for $S$ is upper bounded by $\mathcal{O}\left((\log|S|)^d\right)$. □

# 5 Experiments

## 5.1 Synthetic Experiments

(a) Cosines Function. B=5

(b) Cosines Function. B=10

Figure 1: Immediate regret of the algorithms on the mixture of cosines synthetic function with B = 5 and 10

## 5.2 Real-World Experiments

We first provide the results for the Abalone experiment. We now provide the Prec@1 values for the FastXML

(a) Abalone Function. B=5

(b) Abalone Function. B=10

Figure 2: Immediate regret of the algorithms on the Abalone experiment with B = 5 and 10

experiment on the Delicious experiment.

(a) FastXML - Delicious Dataset. B=5

(b) FastXML - Delicious Dataset. B=10

Figure 3: Immediate regret of the algorithms on the FastXML experiment on the Delicious dataset with B = 5 and 10