[Reviews · NeurIPS 2016]

Reviewer 1

Summary

The paper presents some batch-sequential Bayesian Optimization strategies relying on notions from the theory of Determinantal Point Process. The presented algorithms are related to UCB and to a recently published variant of it, EST. First a Batched-EST (B-EST) is proposed, for which a result is obtained in which the cumulative regret of B-EST is bounded. Second, the DPP-MAX algorithm is presented: it consists, at each batch, in choosing a point by UCB or EST and then exploring a so-called relevance region by maximizing a determinantal criterion, a procedure claimed to be equivalent to "Pure Exploration" in the case where the former criterion is maximized greedily. Third the procedure is randomized by replacing criterion maximization by sampling, hence providing a stochastic algorithm. For these two further algorithms regret bounds are provided as well. Finally some experiments based on analytical functions and realistic test cases are provided and illustrate the good performances of the proposed batch-sequential algorithms in terms of "immediate regret".

Qualitative Assessment

The intersection of good and timely ideas definitely makes of this work an excellent potential NIPS paper. In particular, I have been impressed by the good performances obtained on diverse examples. The theoretical results seem sound even if I am not an expert in this kind of bounds; however I felt a bit disappointed to see that while the actual measure of performance is the so-called "immediate regret" (at the end of the algorithm) what is bounded is the cumulative regret, which does't seem to be a very natural measure of performance in a standard global optimization set-up. In the same flavour, I was surprized not to find references to related works on parallel kriging-based optimization from the computer experiments literature, e.g. on multipoint EI, PI, etc. Also, the mathematical hypotheses are not always clear: is f continuous? Is the domain compact? Also, what if sigma is tending (or equal) to 0? Finally let me mention two points which are not directly studied but might be of interest to investigate here: * What are the speed-ups of the presented batch-sequential strategies compared to their sequential counterparts? * For DPP-SAMPLE, the performances are random. In median or mean they perform better, fine. But what about the risks of failure? In the ideal case random strategies should be good even in terms of higher level quantiles, and the (hence rare) failures should not be too serious; investigating that would be a real plus.

Confidence in this Review

2-Confident (read it all; understood it all reasonably well)


Reviewer 2

Summary

This paper considers the problem of choosing batches of experiments for Bayesian optimization. The approach uses determinantal point processes, in order to choose batches that are diverse. Numerical experiments are provided on both synthetic and real datasets.

Qualitative Assessment

I found the paper interesting and well-written, though it is rather dense. The idea of using DPP for Bayesian optimization is relevant, and the authors provide some practical implementations, as well as reproducible numerical experiments. Another relevant recent reference is: Gonzalez, J., Osborne, M., & Lawrence, N. (2016). GLASSES: Relieving The Myopia Of Bayesian Optimisation. In Proceedings of the 19th International Conference on Artificial Intelligence and Statistics (pp. 790–799).

Confidence in this Review

1-Less confident (might not have understood significant parts)


Reviewer 3

Summary

The problem of optimizing an unknown function is considered, where at each time step, one can sample a batch of B points where to evaluate the function. The problem is considered from a Bayesian standpoint, assuming the unknown function is sampled from a Gaussian process. Borrowing ideas from the rich literature on this topic, the paper proposes a novel algorithm, based on determinantal point processes (DPP). A DPP samples sets of indices proportionally to the determinant of a submatrix formed by this set. The paper introduces several variants, and shows the equivalence between the actions sampled by the algorithm and other strategies. In Section 3.1, a batch version of EST is considered, and is shown to be equivalent to a batch version of UCB. In Section 3.2, Another variant based on DPP sampling, as opposed to maximization is considered and analyzed (Theorem 3.4). Section 4 provides illustrative numerical experiments on a number of problems with both synthetic and real-world datasets.

Qualitative Assessment

Overall, this is an interesting algorithmic contribution, supported by both a regret analysis and extensive numerical experiments. The writing of the paper is not very clear and does not help understanding the exact contributions. The noise is assumed to be exactly Gaussian with known variance \sigma^2. Although this is a standard assumption in the Bayesian world, this is also a restrictive assumption. It would be interesting to study the case when the sigma^2 is unknown, at least. In the literature, there is another body of work related to the algorithms DOO, SOO, STOSOO, POO, etc. that is not cited here. I believe these are relevant works and should be discussed, at least briefly. The contribution should be clarified. First, I think it is important to distinguish better between the existing algorithms from the literature, and the ones that are introduced specifically here. Then, you should clarify the real benefit of using DPP in the end, both in terms of regret performance and numerical efficiency. It seems to me that the key result is Theorem 3.4; perhaps this can be highlighted. Theorem 3.4 is interesting, as it shows some possible improvement one can get using DPP sampling. Do you think you can provide a few hints at the proof in the main material? A natural side question is the computational complexity of the sampling scheme for DPP over other methods, and the effect of possible errors (when using MCMC for instances). The numerical experiments are convincing and reported in a reproducible way. L46: reference? L90: reference?

Confidence in this Review

3-Expert (read the paper in detail, know the area, quite certain of my opinion)


Reviewer 4

Summary

In this paper the authors formulated the Bayesian optimization problem (BBO) as a Bayesian multi-arm bandit problem. Unlike popular methods such as BUCB and GP-UCB where the analysis is based on GPs, they also proposed employing the Determinantal Point Processes (DPPs) to select diverse batches of evaluations and thoroughly analyzed the cumulative regret of the DPP-based method.

Qualitative Assessment

In general the authors have done a good job formulating the problem of BBO into a Bayesian multi-arm bandit problem, and analyzing the proposed method of sampling batches via employing DPPs. I also think that both the algorithmic comparison section (the corresponding parts in introduction, and in section 3.2, 3.3) and experimental section in this paper are comprehensive in comparison to most theoretical MAB papers. My only concern is that there is no matching regret lower bound analysis in the proposed DPP-based algorithm.

Confidence in this Review

1-Less confident (might not have understood significant parts)


Reviewer 5

Summary

The paper describes how to use sampling based DPP-methods to improve Bayesian optimization techniques in the batched case. I really liked the paper for many reasons (see comment section) and recommend it to be accepted to NIPS.

Qualitative Assessment

This was a very interesting paper and I liked reading it a lot. The idea of using DPPs to generate diversity in batches for Bayesain Optimization is pretty cool and I also like that the authors show how sampling-based methods might outperform maximization methods. En passant, they also introduce theoretical bounds for their algorithm and slightly improve upon the known best regret bound for BO so far. Well done. I can only add some minor comments below: 1. Abstract could be a little more concise, at the moment it sounds a bit like parts of the introduction. 2. The introduction sounds a little too broad, I would focus a bit more on Bayesian optimization. 3. The introduction could also shortened a bit, it is a little long at the moment. 4. How complex is the MCMC-based method in run time? The paper currently states it is much faster, but not by what order of magnitude. 5. I'd stress a little more that entropy search is designed for global optimization and not really for the pure regret minimization. It just also happens to perform reasonably well within that scenario, too. 6. Current proof is for radial basis kernel, it would be interesting to see performance for other kernels as well, especially that the Matern kerenl is frequently chosen within BO-settings. 7. I really like section 3.2 btw, the equivalence between PE and DPP maximization is quite intuitive. 8. Sometimes you use the wrong NIPS referencing style. Also, make sure that the references contain capital letters when necessary (for example, Bayes instead of bayes). 9. line 301 "The results for robot were interesting" and line 311 "The results for prec@1" is more interesting; I think these two beginning sound a little odd in comparison.

Confidence in this Review

1-Less confident (might not have understood significant parts)


Reviewer 6

Summary

This paper proposes a new batch Bayesian optimization method involving DPP (determinant point process) maximization/sample. The procedure is close related to UCB-PE or EST (a variant of UCB)-PE. The algorithm selects the first point to maximize UCB type criterion, then the remaining points in the batch is selected by maximizing the diversity. The paper makes the connection between UCB-PE and UCB-DPP-MAX by realizing that the second stage in the UCB-PE is just maximizing DPP via a greedy algorithm. Then naturally authors propose the DPP-SAMPLE to replace directly maximizing DPP inspired by the previous work on DPP. The paper is solid in both theory and experiments. The paper generalizes/improves the cumulative regret bound, and also defends the method empirically.

Qualitative Assessment

At the high level, the paper makes the follows contributions: (1) It connects EST with UCB, which naturally UCB-PE to EST-PE. (2) It connects UCB-PE to the DPP maximization by a greedy algorithm. (3) It proposes DPP-Sample for Batch Bayesian optimization and proves a better regret bound. (4) The paper conducts several experiments, demonstrating the effectiveness of the algorithm in practice. In details, I would like to give the following suggestions: (1) Can authors show the error bar/ confidence interval besides the median in the experiments (2) Can the authors compare to the parallel EI available in spearmint or MOE? All in all, I think this paper is of good quality, and meets the standard of NIPS publication.

Confidence in this Review

3-Expert (read the paper in detail, know the area, quite certain of my opinion)